# MMPOI: A Multi-Modal Content-Aware Framework for POI Recommendations

## ABSTRACT

The Point-of-Interest (POI) recommendation system, designed to recommend potential future visits of users based on their check-in sequences, faces the challenge of data scarcity. This challenge primarily stems from the data sparsity issue, namely users interact with only a small number of POIs. Most existing studies attempt to solve this problem by focusing on POI check-in sequences, without considering the substantial multi-modal content information (e.g. textual and image data) commonly associated with POIs. In this paper, we propose a novel multi-modal content-aware framework for POI recommendation (MMPOI). Our approach addresses the issue of data sparsity by incorporating multi-modal content information about POIs from a new perspective. Specifically, MM-POI leverages pre-trained models for inter-modal conversion and employs a unified pre-trained model to extract modal-specific features from each modality, effectively bridging the semantic gap between different modalities. We propose to build a Multi-Modal Trajectory Flow Graph (MTFG) which combines the multi-modal semantic structure with check-in sequences. Moreover, we design an adaptive multi-task Transformer that models users' multi-modal movement patterns and integrates them for the next POI recommendation tasks. Extensive experiments on four real-world datasets demonstrate that MMPOI outperforms state-of-the-art POI recommendation methods. To facilitate reproducibility, we have released both the code and the multi-modal POI recommendation datasets we collect[1].

## CCS CONCEPTS

• **Information systems → Recommender systems**.

## KEYWORDS

Multi-modal, POI recommendation, Recommender systems

**ACM Reference Format:**
. 2018. MMPOI: A Multi-Modal Content-Aware Framework for POI Recommendations. In *Proceedings of Make sure to enter the correct conference title from your rights confirmation emai (Conference acronym 'XX).* ACM, New York, NY, USA, 10 pages. https://doi.org/XXXXXXX.XXXXXXX

## 1 INTRODUCTION

POI recommender systems are designed to capture the preference of users based on their current and historical footprints, also known

[1]https://github.com/**blind**/MMPOI

2023-10-11 15:20. Page 1 of 1–10.

as check-ins, and many recent studies [19, 32, 41, 42] on POI recommendations aim to predict the next POI that users will be interested in. POI recommendation not only provides individuals with a convenient exploration of unfamiliar locales but also empowers businesses with sharpened marketing tactics [39]. The primary supervisory signals used for training these model parameters typically originate from interactions between users and POIs, manifested as user check-in sequences. However, the sparsity issue arises as users often visit only a few preferred POIs, which constitute a very small portion of the entire POI database [16, 18]. Given the sparsity of user check-in data, POI recommendation methods that solely rely on such data can be heavily impacted by the data sparsity problem, making it challenging to accurately predict the next POI a user would visit.

The existing methods tend to incorporate side information such as temporal information [20, 42, 45], geographical locations [32, 40], categories [41, 44], and social relationships [6, 10, 21, 26] to alleviate the issue of data sparsity. Some studies also utilize hypergraphs [30, 38] and knowledge graphs [24, 31] to explore higher-order user-POI relationships, or employ sampling techniques [4, 17] to mitigate the challenge of sparse data. While these methods have made considerable advancements, none of them consider the significant amount of multi-modal content information associated with POIs (e.g., visual and textual content).

Inspired by the achievements of multi-modal recommendation methods [43, 46] and pre-trained language models [8], we propose to apply pre-trained models to effectively exploit the multi-modal content information of POIs, and integrate such multi-modal information into check-in sequences to enhance the accuracy of POI recommendation. However, incorporating the multi-modal content information of POIs with existing POI recommendation frameworks presents three noteworthy challenges. Firstly, there exist substantial variations in the semantic spaces of different modal content. For instance, the visual and textual contents of POIs are typically represented in different semantic spaces, posing a challenge in merging multi-modal features. Secondly, multi-modal data contains a considerable volume of noise, which could introduce a significant quantity of noise into POI recommendation tasks, thereby detrimentally impacting the accuracy of recommendation. Thirdly, there is a significant semantic difference between the multi-modal content of POIs and user check-in sequences. The multi-modal contents describe the common characteristics of POIs, while the check-in sequences reflect the interaction behavior of users. These two types of data have substantial semantic differences. The challenge lies in how to effectively integrate these two types of data to model user movement patterns.

To this end, we propose a novel Multi-Modal content-aware framework for POI recommendation (MMPOI). To address the first challenge, namely the different representation spaces between textual and visual modalities, MMPOI employs an image2text pre-trained model to convert POI images into natural language descriptions. Subsequently, a unified pre-trained language model is

applied to extract features from each modality, thereby mapping multi-modal content to a shared semantic representation space. To address the second challenge of considerable noise intruding from multi-modal content, we establish modal-specific similarity structure graphs to model the latent semantic correlation of POIs, and adopt the $k$NN sparsification method to filter out the important relationships. This strategy effectively mitigates the impact of multi-modal noise on the accuracy of recommendations. In order to address the third challenge brought by the semantic difference between multi-modal content and user check-in behavior, we propose a Multi-modal Trajectory Flow Graph (MTFG) that integrates the multi-modal latent semantic relationships with check-in sequences. Additionally, we construct a Geographic Trajectory Flow Graph (GTFG) to capture the geographical sequence relationships. Finally, we design an adaptive multi-task Transformer by taking into account various factors that influence user behavior, which is employed to model the user's comprehensive movement patterns for the next POI recommendation. The primary contributions of this paper can be summarized as follows:

- We propose a novel MMPOI model to address the issue of data sparsity by incorporating the multi-modal content information of POIs. To the best of our knowledge, our study represents the first attempt to utilize multi-modal content information of POIs for the next POI recommendation.
- MMPOI leverages pre-trained models to map multi-modal content to a shared semantic space, and constructs a multi-modal trajectory flow graph to effectively integrate de-noised multi-modal knowledge with check-in sequences. Moreover, MMPOI establishes a geographic trajectory flow graph to extract geographical sequence patterns and employs an adaptive multi-task Transformer to capture users' comprehensive movement patterns.
- To support multi-modal POI recommendation, we collected mult-modal content for the widely used Foursquare dataset. Experiments conducted on Foursquare and Yelp datasets show that MMPOI can outperform the strongest baseline by 8% to 11% in recommendation accuracy.

## 2 RELATED WORK

### 2.1 Next POI Recommendation

Next POI recommendation aims to predict the users' next moves based on users' historical check-in data. Early work is built upon the Markov chains and focuses on predicting users' preferences between POIs to recommend the next visit [5, 7]. Recent work often employs recurrent neural networks (RNN) and self-attention mechanisms. For example, LSTPM [27] uses contextual information of POIs to model users' long-term preferences and a geo-dilated RNN to model short-term preferences. STAN [22] learns the explicit spatio-temporal correlations within the user trajectory using a bi-attention architecture. These methods focus on capturing the spatio-temporal relationships between POIs in a single check-in sequence, and the relationship among multiple check-in sequences is not well exploited. To address this, some recent studies incorporate graph representation learning techniques. GETNext [39] constructs a directed trajectory graph to represent the correlation

between multiple check-in sequences, and applies Graph Convolutional Network (GCN) to learn the representations of POIs. AGRAN [32] combines geographical dependencies learned from the adaptive graph and spatio-temporal information simultaneously for capturing dynamic user preferences. STHGCN [38] introduces a hypergraph to learn the trajectory-grain information from the user's historical trajectories and collaborative trajectories from other users. However, existing POI recommendation methods ignore the multi-modal content information associated with POIs. In this paper, we introduce a multi-modal content-aware framework for POI recommendation, marking the first attempt to exploit the multi-modal content information for POI recommendations.

### 2.2 Multi-Modal Recommendation

Multi-modal recommendation methods exploit massive multi-modal content information of items to improve recommendation performance, which have been successfully applied to many applications, such as micro-video platforms, social media platforms, and e-commerce [43]. Early approaches tend to integrate the multi-modal content features of items into matrix factorization frameworks for recommendation [2, 9, 37]. For example, VBPR [9] extends the matrix factorization by extracting image features from item images to improve the recommendation performance. VECF [2] introduces a multi-modal attention network to capture users' multiple interests across both image regions and reviews. Recently, an increasing number of studies have incorporated GNNs to model multi-modal features [34, 35, 43]. For example, MMGCN [35] utilizes modal-specific graphs and graph convolutional operations to capture both modal-specific user preferences and item representations. LATTICE [43] combines multi-modal features to identify item-item graph structures and integrates these multi-modal similarity relationships with the traditional collaborative filtering method. These methods model the latent semantic relationships between items based on GCN, but cannot be directly applied to spatio-temporal data. Additionally, some studies introduce contrastive learning methods to fuse multi-modal features for recommendation [28, 33, 46]. For example, BM3 [46] utilizes a node discarding mechanism to perturb user and item embeddings, and introduces a multi-modal contrastive learning paradigm to align feature representations across different modalities. MMSSL [33] specializes in modal-specific user and item embeddings through adversarial perturbation and introduces cross-modal contrastive learning for capturing inter-modality interaction dependencies. Different from these multi-modal recommendation methods, we utilize pre-trained models to map multi-modal data into a shared semantic space. Then, we use the k-nearest neighbors sparsification approach to filter out noise in the multi-modal data. Moreover, we effectively integrate multi-modal content features of POIs with check-in sequences to address the POI recommendation task.

## 3 THE PROPOSED METHOD

In this section, we first formulate the problem and introduce our proposed MMPOI framework. As shown in Figure 1, there are four main components in our proposed MMPOI framework: (1) An image2text pre-trained model is utilized to convert POI images into textual descriptions, and then a unified pre-trained language model is employed to extract modal-specific features from each modality.

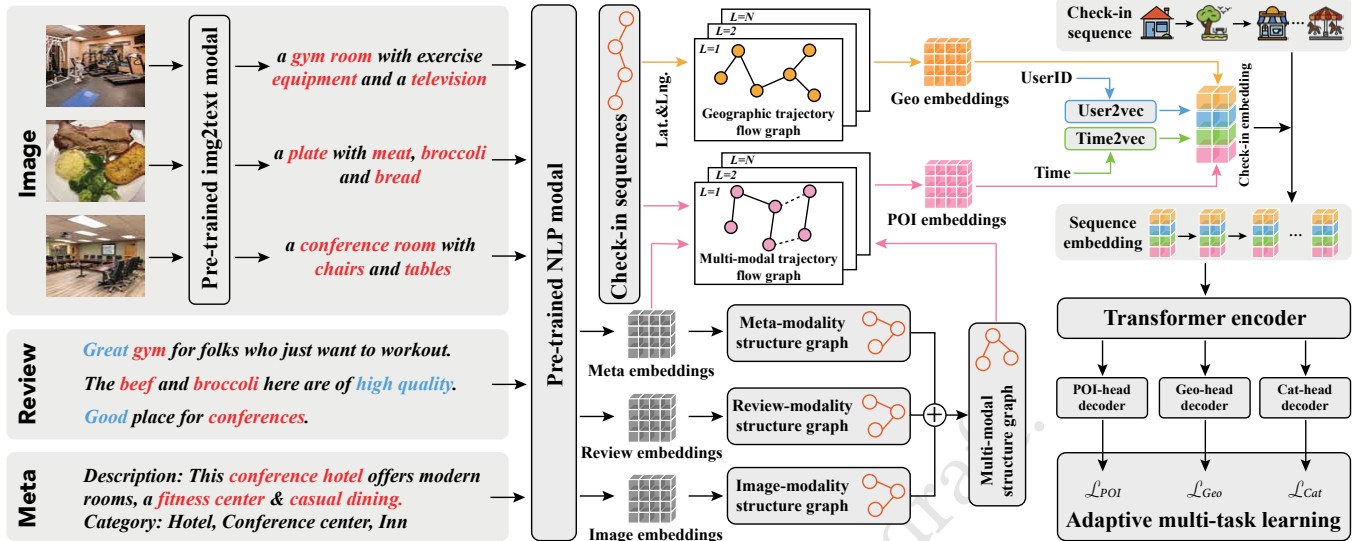

**Figure 1: The model flow of our proposed MMPOI.**

(2) Modal-specific features of POIs are aggregated and combined with user check-in sequences, constructing a multi-modal trajectory flow graph for POI representation learning. (3) A geographic trajectory flow graph is established to model geographical sequence relationships. Check-in representations are learned from spatial location, multi-modal POI content, user preferences, and temporal patterns. (4) An adaptive multi-task Transformer is proposed to model the user movement patterns and provide POI recommendations.

### 3.1 Preliminaries

Let $P = \{p_1, p_2, ..., p_N\}$ be the set of $N$ POIs and $U = \{u_1, u_2, ..., u_M\}$ be the set of $M$ users. Each user $u \in U$ has a check-in sequence $C_u = \{c_u^1, c_u^2, c_u^3, ...\}$, where a check-in is denoted as a tuple $c = (u, p, t)$, indicating that the user $u$ visits POI $p$ at time $t$. The check-in sequences of all users can be represented as $C = \{C_{u_1}, C_{u_2}, ..., C_{u_M}\}$. Besides user-POI check-ins, we also innovatively consider the multi-modal content information of POIs. Each POI $p \in P$ is associated with a set of multi-modal content $(\mathcal{V}_p, \mathcal{R}_p, \mathcal{M}_p)$, where $\mathcal{V}_p = \{v_p^1, v_p^2, ..., v_p^{|\mathcal{V}_p|}\}, \mathcal{R}_p = \{r_p^1, r_p^2, ..., r_p^{|\mathcal{R}_p|}\}$ and $\mathcal{M}_p = \{m_p^1, m_p^2, ..., m_p^{|\mathcal{M}_p|}\}$ denote sets of images, reviews, and metadata for the POI $p$, and $|\mathcal{V}_p|, |\mathcal{R}_p|$ and $|\mathcal{M}_p|$ represent the number of images, reviews, and metadata, respectively. Then, each POI $p \in P$ can be denoted by a tuple $p = (lat_p, lon_p, cat_p, \mathcal{V}_p, \mathcal{R}_p, \mathcal{M}_p)$ of latitude, longitude, category, and multi-modal contents, respectively. Given the check-in sequence $C_u = \{c_u^1, c_u^2, ..., c_u^{|C_u|}\}$ of the target user $u$, the goal of POI recommendation is to predict the most likely future POIs that user $u$ would visit next.

### 3.2 Multi-Modal Feature Extraction

As shown in Figure 1, our proposed method considers content information from three POI modalities: images, reviews, and metadata. To comprehensively utilize multi-modal data, it's crucial to address the semantic space differences between different modalities, especially between the visual and textual modalities. We employ a

pre-trained image2text model BLIP2 [15] to convert the image of POIs into the corresponding textual descriptions:

$$\mathcal{V}_p' = \{v_p^{1'}, v_p^{2'}, ..., v_p^{|\mathcal{V}_p|'}\}$$
$$= \{BLIP2(v_p^1), BLIP2(v_p^2), ..., BLIP2(v_p^{|\mathcal{V}_p|})\}. \quad (1)$$

From the example presented in Figure 1, we can see that the pre-trained BLIP2 model provides natural language descriptions of entities and their relationships present in POI images, while making inferences about the image scenes. Intuitively, we can observe a clear semantic consistency across different modalities of the same POI after modality conversion. Then, we employ a unified pre-trained language model Sentence-BERT [25] to extract modal-specific features from each modality. Specifically, considering that a POI can be associated with multiple images, we obtain the image-modality representation of POI $p$ by feeding $v_p^{i'} \in \mathcal{V}_p'$ into Sentence-BERT and averaging the embedding results:

$$e_p^v = mean(sum(BERT(v_p^{1'}), BERT(v_p^{2'}), ..., BERT(v_p^{|\mathcal{V}_p|'}))), \quad (2)$$

where $e_p^v \in \mathbb{R}^{d_m}$ is POI $p$'s image-modality representation, $d_m$ represents the dimension of the feature representation.

Similar to the image modality, the review-modality representation of POI $p$ can be obtained as follows:

$$e_p^r = mean(sum(BERT(r_p^1), BERT(r_p^2), ..., BERT(r_p^{|\mathcal{R}_p|}))). \quad (3)$$

The metadata we primarily consider includes descriptions and categories of POIs. As the metadata holds explicit semantics, we extract semantic features from metadata as its meta-modality representation:

$$e_p^m = mean(sum(BERT(m_p^1), BERT(m_p^2), ..., BERT(m_p^{|\mathcal{M}_p|}))) \quad (4)$$

### 3.3 Multi-modal Trajectory Flow Graph

Existing POI recommendation methods typically rely on modeling user check-in sequences to predict future user behaviors [22, 36, 39]. However, user check-in sequences are very sparse, which presents

substantial challenges for model learning. In this paper, we aim to leverage the multi-modal content information of POIs to alleviate the data sparsity issue and improve recommendation performance. Nevertheless, directly integrating multi-modal features inevitably introduces a lot of noise. Therefore, inspired by [43], we construct a modal-specific structure graph for each modality, which captures modal-specific $k$-nearest-neighbor ($k$NN) relationships between POIs. This method serves to filter out noise and capture important structural relationships among the latent features of POIs. To effectively integrate the multi-modal features of POIs and the check-in sequence features of users, we construct a denoised dense multi-modal trajectory flow graph (MTFG) to learn the multi-modal representations of POIs. The following sections will provide a detailed introduction to the MTFG construction process.

### 3.3.1 Learning Modal-Specific Structure Graph.
Given the modal-specific representations $\{e_p^o\}_{p \in P}$ for POIs in modality $o$, where $o \in O$, and $O = \{\mathcal{V}, \mathcal{R}, \mathcal{M}\}$ represents the set of modalities. We compute the similarity score $S_{ij}^o$ of POI pairs $(i, j)$ by employing a cosine similarity function on their modal-specific representations $e_i^o$ and $e_j^o$ as follows:

$$S_{ij}^o = \frac{(e_i^o)^\top e_j^o}{\| e_i^o \| \| e_j^o \|}. \qquad (5)$$

We denote the similarity matrix among $N$ POIs in modality $o$ as $S^o \in \mathbb{R}^{N \times N}$, where $S_{ij}^o$ is the element at the $i$-th row, $j$-th column. $S^o$ represents a homogeneous graph that characterizes the similarity structures among POIs within modality $o$. However, considering the substantial noise present in multi-modal data and the computational and storage costs associated with a fully connected graph, we employ $k$NN sparsification [1] on $S^o$. For each POI node $p_i$, we only keep edges with the top-$k$ similarity scores:

$$\widehat{S}_{ij}^o = \begin{cases} S_{ij}^o, & S_{ij}^o \in top\text{-}k(S_i^o) \\ 0, & otherwise, \end{cases} \qquad (6)$$

where $\widehat{S}^o = \{\widehat{S}_{ij}^o\}_{i,j \in [1,N]}$ is the sparsified graph adjacency matrix, representing the denoised similarity structure among POIs within modality $o$. In this paper, we use $\widehat{S}^o$ to represent the $o$-modality structure graph.

### 3.3.2 Aggregating Multi-Modal Structure Graph.
In POI recommendation scenarios, user behavior is significantly influenced by multi-modal content information. For instance, when choosing a restaurant, users typically first consider the restaurant's category (meta) and then browse images of the restaurant's ambiance and dishes (image) to form sensory evaluations. Subsequently, they might read other customers' critiques of the restaurant (review) to evade potential dissatisfaction and needless overspending. To model the comprehensive influence of multi-modal factors, we combine the obtained structure graph $\widehat{S}^o$ for each modality $o \in O$ to create a multi-modal structure graph $S^M$:

$$S^M = \sum_{o=1}^{|O|} \widehat{S}^o, \qquad (7)$$

where $S^M \in \mathbb{R}^{N \times N}$ denotes the multi-modal structure graph that reflects multi-modal semantic relationships of POIs. In particular,

to mitigate the issue of exploding or vanishing gradients [14], we normalize $S^M$ as follows:

$$\widetilde{S^M} = D^{-\frac{1}{2}} S^M D^{-\frac{1}{2}}, \qquad (8)$$

where $D \in \mathbb{R}^{N \times N}$ is the diagonal degree matrix of $S^M$ and $D_{ii} = \sum_j S_{ij}^M$.

### 3.3.3 Building User Trajectory Graph.
Considering that similar movement patterns may exist among different users and the same user may repeat certain historical behaviors multiple times. To model the similarity relationships among these similar sequence segments, we create a directed weighted user trajectory graph.

To be specific, given the set of user check-in sequences $C = \{C_u\}_{u \in U}$, the user trajectory graph can be defined as a directed weighted graph $\mathcal{G}^T = (V^T, E^T, w)$. $V^T = P$ is the set of all POI nodes. $E^T$ represents the set of directed edges $(p_i, p_j)$, which indicates that POI $p_j$ appears after POI $p_i$ in some user check-in sequences, i.e. they are visited consecutively. The adjacency matrix of $\mathcal{G}^T$ can be denoted as $S^T = \{S_{ij}^T\}_{i,j \in [1,N]}$, where

$$S_{ij}^T = \begin{cases} w(p_i, p_j), & (p_i, p_j) \in C \\ 0, & otherwise, \end{cases} \qquad (9)$$

where $w(p_i, p_j)$ equals the number of occurrences of edge $(p_i, p_j)$ in user check-in sequences $C$. Similar to the multi-modal structure graph $S^M$, we normalize the user trajectory graph $S^T$ as:

$$\widetilde{S^T} = (D^t)^{-\frac{1}{2}} S^T (D^t)^{-\frac{1}{2}}, \qquad (10)$$

where $D^t \in \mathbb{R}^{N \times N}$ is the diagonal degree matrix of $S^T$ and $D_{ii}^t = \sum_j S_{ij}^T$.

### 3.3.4 Multi-modal Trajectory Flow Graph (MTFG).
To bridge the semantic difference between POI multi-modal features and check-in sequence features, we integrate the multi-modal structure graph with the user trajectory graph, creating the multi-modal trajectory flow graph $\mathcal{G}^{MT}$ as follows:

$$S^{MT} = \alpha \widetilde{S^M} + (1 - \alpha) \widetilde{S^T}, \qquad (11)$$

where $S^{MT}$ is the adjacency matrix of $\mathcal{G}^{MT}$, $\alpha$ is a hyper-parameter used to integrate the multi-modal structural relationships and the check-in sequence relationships between POIs. It's noteworthy that the MTFG we constructed is a mixed graph with both directed and undirected edges, representing the two types of relationships previously discussed. Due to the $k$NN sparsification employed in multi-modal structure graphs, noise can be considerably mitigated. Consequently, the constructed MTFG not only addresses the data sparsity issue but also emphasizes multiple key relationships between POIs.

### 3.3.5 Multi-Modal POI Representation Learning.
To take full advantage of the topological structure of MTFG, the spectral GCN method [14] is adopted in MTFG to learn multi-modal POI representations. Specifically, the multi-modal POI representations $E_P^{(l+1)}$ at the $(l + 1)$-th graph convolution layer is obtained as follows:

$$E_P^{(l+1)} = \sigma(S^{MT} E_P^{(l)} W_P^{(l+1)} + b_P^{(l+1)}), \qquad (12)$$

where $\sigma(\cdot)$ denotes the leaky ReLU activation function, $W_P^{(l+1)} \in \mathbb{R}^{d^{(l)} \times d^{(l+1)}}$ is the weight matrix at the $(l + 1)$-th layer, $b_P^{(l+1)} \in$

$\mathbb{R}^{d^{(l+1)}}$ represents the corresponding bias, and $d^{(l)}$ is the number of channels in the $l$-th hidden layer.

In particular, considering that the 'category' attribute in the meta modality carries concise and explicit semantic information, we utilize its semantic representation, denoted as $E_C \in \mathbb{R}^{N \times d_m}$, to serve as the feature matrix for the input nodes, i.e., $E_P^{(0)} = E_C$. Details about the embedding of categories will be provided in the following sections. Then, assuming GCN has a total of $L$ layers, to enhance the model's generalization ability, we apply random dropout to the features before the final layer. Using this approach, we can formulate the output of the GCN as follows:

$$E_P = S^{MT} Dropout(E_P^{(L-1)}, \theta_{GCN}) W_P^{(L)} + b_P^{(L)}, \tag{13}$$

where $E_P = \{e_p^1, e_p^2, ..., e_p^N\} \in \mathbb{R}^{N \times d}$ denotes the multi-modal POI representations, $Dropout(\cdot)$ is an element-level dropout operation applied to the feature representations, and $\theta_{GCN}$ represents its dropout ratio. The learned multi-modal POI representations integrate both the multi-modal semantic relationships and the check-in sequence patterns between POIs. This contributes to a more comprehensive comprehension of user movement patterns in the POI recommendation task.

## 3.4 Check-in Representation Learning

Many POI recommendation methods have demonstrated the significance of considering the check-in context in predicting user check-in behaviors [24, 32, 38]. In this section, we propose a check-in representation learning approach that takes into account factors such as spatial location, temporal information, multi-modal POI features, and user preferences.

*3.4.1 Geographic Trajectory Flow Graph.* The check-in behavior of users exhibits a strong regional dependence [19]. For example, people usually prefer shopping at a local grocery store instead of traveling a long distance to a remote one. To effectively leverage the geographical information, we propose a Geographic Trajectory Flow Graph (GTFG) to model geographical relationships in user movement patterns. To be specific, we apply the publicly available geocoding system Geohash[2] to encode the spatial location of POIs. Given a POI $p$ with location coordinates $(p_{lat}, p_{lng})$, $G@Z$ maps $(p_{lat}, p_{lng})$ to the respective grid cell $p^G$. These grid cells are of equal size, and the precision parameter $Z$ determines the size of these fixed cells. As the value of $Z$ increases, the size of these fixed cells diminishes. To learn fine-grained spatial relationships, we set the value of $Z$ to 6 in our method. Given the set of user historical check-in sequences $C$, GTFG can be denoted as $\mathcal{G}^G = (V^G, E^G, w)$, where $V^G = \{p_1^G, p_2^G, ..., p_{|V^G|}^G\}$ represents the set of grid cells of all POIs. $E^G$ is the set of directed edges $(p_i^G, p_j^G)$, which indicates that the user visited the grid cell $p_j^G$ after visiting the grid cell $p_i^G$. The adjacency matrix of $\mathcal{G}^G$ can be represented as:

$$S^G = \{S_{ij}^G\} = \begin{cases} w(p_i^G, p_j^G), & (p_i^G, p_j^G) \in C \\ 0, & otherwise, \end{cases} \tag{14}$$

where $w(p_i^G, p_j^G)$ represents the number of times edge $(p_i^G, p_j^G)$ appears in $C$. We normalize the $S^G$ as follows:

$$\widetilde{S^G} = (D^G)^{-\frac{1}{2}} S^G (D^G)^{-\frac{1}{2}}, \tag{15}$$

where $D^G \in \mathbb{R}^{N \times N}$ is the diagonal degree matrix of $S^G$ and $D_{ii}^G = \sum_j S_{ij}^G$. Similar to the multi-modal POI representation learning, we apply the spectral GCN method to the constructed GTFG to learn the geographic representations. The graph convolution process and the GCN output can be represented as follows:

$$E_G^{(l+1)} = \sigma(\widetilde{S^G} E_G^{(l)} W_G^{(l+1)} + b_G^{(l+1)}), \tag{16}$$

$$E_G = \widetilde{S^G} Dropout(E_G^{(L-1)}, \theta_{GCN}) W_G^{(L)} + b_G^{(L)}, \tag{17}$$

where $E_G = \{e_g^1, e_g^2, ..., e_g^N\} \in \mathbb{R}^{N \times d}$ is geographic representations of all POIs, $W_G^{(l+1)}$ and $b_G^{(l+1)}$ are the weight matrix and bias term at the $(l+1)$-th layer, respectively. We apply one-hot encoding to the grid cells to initialize the geographic representation $E_G^{(0)}$.

*3.4.2 Time and Category Embedding.* Previous studies on user movement patterns have demonstrated a significant correlation between the time of user visits and the category of POIs visited. [39]. For instance, users typically visit bars and similar venues during the nighttime, whereas subway stations are busier during peak commuting hours. Therefore, to model the relationship between user movement patterns and both time and categories, we encode visiting time and POI categories separately.

For the visiting time, we partition the 24-hour day into 48 slots, each accounting for 30 minutes. We then apply the time2vector [12] method to embed these slots, thereby obtaining the time embedding representations $E_T = \{e_t^1, e_t^2, ..., e_t^{48}\} \in \mathbb{R}^{48 \times d}$.

For category information, we take into account the rich semantics within categories. For example, consider the following four categories: "French restaurant", "Chinese restaurant", "restaurant", and "dining establishment". Each of these categories holds a distinct textual description yet they also share notable similarities. Using one-hot encoding for these categories would treat them as completely separate entities. On the other hand, simplifying them under a general "restaurant" category could ignore the important and nuanced distinctions such as "Chinese" or "French". In this paper, we consider categories as textual content and use the pre-trained natural language model Sentence-BERT [25] to obtain the representations of all POIs' categories, denoted as $E_C = \{e_c^1, e_c^2, ..., e_c^N\} \in \mathbb{R}^{N \times d_m}$. It's worth noting that the 'category' has been completely incorporated into the POI learning process, where it's deeply integrated with multi-modal POI representations. Thus, we no longer need to explicitly incorporate category features into the check-in representation.

*3.4.3 Check-in Representation.* We have taken into account factors like multi-modal content, spatial location, and temporal information. However, user check-in behavior, an active interaction with POIs, may be impacted by multiple potential factors. To manage these, we train an embedding layer $f(\cdot)$ to generate a $d$-dimensional embedding for each user $u$:

$$e_u = f(u) \in \mathbb{R}^d. \tag{18}$$

---

[2]http://geohash.org/, where G@Z= 2 (1,251km × 625km), G@Z= 3 (156km × 156km), G@Z= 4 (39km × 19.5km), G@Z= 5 (4.9km × 4.9km), G@Z= 6 (1.2km × 0.61km).

Then, we can represent user $u$'s check-in record $c_u^i$ by concatenating the corresponding factor embeddings:

$$x_u = f_{embed}(c_u^i) = Concat(e_u, e_p, e_g, e_t), \quad (19)$$

where $x_u \in \mathbb{R}^{4d}$ represents check-in embedding, $e_u$ is user $u$'s embedding, $e_p$ is the visited POI's multi-modal representation, $e_g$ denotes the embedding of the grid cell where the check-in location is situated, and $e_t$ represents the embedding of the check-in time.

## 3.5 Adaptive Multi-task Transformer

*3.5.1 Transformer Encoder.* To limit the risk of over-smoothing, GCN is generally kept to a lower depth, which means that only local features in check-in sequences are captured using GCN-based methods. Therefore, we adopt a Transformer encoder to model global user movement patterns. Given a user check-in sequence $C_u = \{c_u^1, c_u^2, ..., c_u^{l_m}\}$ containing up to $l_m$ check-ins, the corresponding check-in embeddings are then systematically stacked to form an input tensor $\mathcal{X}_u = \{x_u^1, x_u^2, ..., x_u^{l_m}\}$ of the first encoder layer. Considering that an accurate prediction of the next POI category and grid cell could improve the model's comprehension of user preferences and movement patterns. In addition to the main task of predicting the next POI, we develop several auxiliary tasks to aid in the training of our proposed MMPOI. At the decoding stage, we constructed a multi-head decoder based on Multi-Layer Perceptron (MLP) to simultaneously perform multiple prediction tasks:

$$\hat{Y}_{poi} = \mathcal{X}^{output} W_{poi} + b_{poi}, \quad (20)$$

$$\hat{Y}_{geo} = \mathcal{X}^{output} W_{geo} + b_{geo}, \quad (21)$$

$$\hat{Y}_{cat} = \mathcal{X}^{output} W_{cat} + b_{cat}, \quad (22)$$

where $W_{poi} \in \mathbb{R}^{4d \times N}$, $W_{geo} \in \mathbb{R}^{4d \times |V^G|}$, and $W_{cat} \in \mathbb{R}^{4d \times d}$ are weights of MLPs, $\mathcal{X}^{output}$ represents the output of the Transformer encoder, and $|V^G|$ is the number of grid cells. $\hat{Y}_{poi}$, $\hat{Y}_{geo}$ and $\hat{Y}_{cat}$ are respectively the model's predictions for the POI, the grid cell, and the category for the next check-in.

*3.5.2 Adaptive Multi-task Learning.* During model training, we simultaneously consider the prediction loss from multiple decoders. Cross entropy is utilized as the loss function for both the next POI and the next grid cell prediction tasks. Furthermore, Kullback-Leibler divergence serves as an evaluation metric for category prediction performance. To balance the weights of multiple losses, most POI recommendation methods tend to sum up losses for each individual task with weighted linear combinations, which typically necessitates manual weight adjustments. Nonetheless, the model performance is greatly influenced by these parameters, and fine-tuning these parameters manually is a time-consuming and challenging task in practice.

To this end, we introduce a multi-task learning method based on task-dependent uncertainty [13], which effectively trains the model through adaptively adjusted multi-task weights. Finally, the overall loss function of the proposed model can be written as:

$$\mathcal{L} = \frac{1}{2\sigma_1^2}\mathcal{L}_{poi} + \frac{1}{2\sigma_2^2}\mathcal{L}_{cat} + \frac{1}{2\sigma_3^2}\mathcal{L}_{geo} + log\sigma_1\sigma_2\sigma_3, \quad (23)$$

where $\sigma_1$, $\sigma_2$, and $\sigma_3$ denote learnable uncertainty in three prediction tasks, respectively. It can be deduced that the larger $\sigma$, the higher the uncertainty of the task, and consequently, the smaller the weight assigned to that specific task. This implies that during

**Table 1: Statistics of the datasets. #Seq. denotes the number of check-in sequences.**

| Dataset | #User | #POI | #Check-in | #Seq. | Density |
|---|---|---|---|---|---|
| NYC | 1,080 | 4,637 | 115,134 | 32,546 | 0.01016 |
| TKY | 2,291 | 7,219 | 387,304 | 90,242 | 0.00286 |
| New_orleans | 1,011 | 2,816 | 44,821 | 2,188 | 0.01574 |
| Philadelphia | 3,315 | 6,563 | 147,152 | 7,418 | 0.00676 |

the training process, the model will prioritize learning simple tasks that have less noise and are easier to optimize. However, it's important to note that noisier and more challenging tasks are not disregarded. We will explore the impact of introducing multi-task uncertainty on model learning through our experiments.

# 4 EXPERIMENTS

## 4.1 Experimental Settings

*4.1.1 Datasets.* We conduct experiments on four public datasets: FourSquare-NYC[3], FourSquare-TKY[3], Yelp-New_orleans[4], and Yelp-Philadelphia[4]. The two Yelp datasets are obtained by splitting the original Yelp dataset according to cities. For all datasets, we first perform 10-core filtering, which means removing unpopular POIs and low-interaction users with less than 10 check-in records. Then, we collect multi-modal content information such as images and reviews for POIs to ensure that every POI in the dataset contains content information from three modalities: image, review, and metadata (description or/and category). Moreover, we split the user's check-in sequence at intervals of 24 hours. In particular, we filter out sequences with only 1 check-in after splitting. Finally, we divide the datasets into training and testing sets with a ratio of 8:2. Table 1 presents statistics for these datasets.

*4.1.2 Evaluation Metrics.* To evaluate the performance of the next POI prediction, we adopt two common ranking evaluation methods, Hit Ratio (HR) and Normalized Discounted Cumulative Gain (NDCG), to evaluate the quality of the recommendation list. HR@K is a metric commonly used in previous POI recommender systems [19, 39] to determine whether the target user's next check-in location appears within the top-K recommendation list. NDCG@K is a widely accepted measure for evaluating recommendation algorithms [29, 43]. It takes into account both ranking precision and the position of ratings in its evaluation.

*4.1.3 Baselines.* To evaluate the performance of our proposed model MMPOI, we select three categories of methods as baselines: multi-modal recommendation methods (**LATTICE** [43], **MMSSL** [33]), sequential recommendation methods (**SASRec** [11], **ICLR** [3]), and POI recommendation methods (**LSTPM** [27], **STAN** [22], **PLSPL** [36], **GETNext** [39], **DisenPOI** [23]). Detailed descriptions of these methods will be provided in the appendix.

*4.1.4 Implementation Details.* We implement our proposed model by PyTorch, and set the feature embedding dimension $d$ to 128. We utilize Adam as the optimizer and fix the batch size to 64 for all models. For a fair comparison, we carefully tune the parameters of each model following their published papers. As we employ

---

[3] https://sites.google.com/site/yangdingqi/home/foursquare-dataset
[4] https://www.yelp.com/dataset

**Table 2: Performance of baselines in terms of HR@k and NDCG@k on four datasets.**

| Method | NYC | | | TKY | | | New_orleans | | | Philadelphia | | |
|---|---|---|---|---|---|---|---|---|---|---|---|---|
| | HR@5 | HR@20 | NG@20 | HR@5 | HR@20 | NG@20 | HR@5 | HR@20 | NG@20 | HR@5 | HR@20 | NG@20 |
| SASRec [11] | 0.34724 | 0.46892 | 0.29420 | 0.19345 | 0.28738 | 0.17051 | 0.02229 | 0.06834 | 0.02209 | 0.01221 | 0.03891 | 0.01520 |
| ICLR [3] | 0.36540 | 0.49338 | 0.30462 | 0.21948 | 0.32758 | 0.20495 | 0.02572 | 0.07633 | 0.02638 | 0.01563 | 0.04934 | 0.01877 |
| LATTICE [43] | 0.32182 | 0.38071 | 0.27353 | 0.16704 | 0.26360 | 0.15844 | 0.02091 | 0.06065 | 0.02044 | 0.01098 | 0.03574 | 0.01412 |
| MMSSL [33] | 0.33763 | 0.45260 | 0.28437 | 0.18504 | 0.27463 | 0.16578 | 0.02172 | 0.06645 | 0.02145 | 0.01193 | 0.03809 | 0.01489 |
| LSTPM [27] | 0.35654 | 0.47823 | 0.29819 | 0.20310 | 0.30176 | 0.17870 | 0.02360 | 0.07097 | 0.02308 | 0.01315 | 0.04170 | 0.01628 |
| PLSPL [36] | 0.40702 | 0.52563 | 0.32011 | 0.21532 | 0.32192 | 0.18856 | 0.02463 | 0.07420 | 0.02383 | 0.01337 | 0.04246 | 0.01652 |
| STAN [22] | 0.45969 | 0.59120 | 0.35932 | 0.24183 | 0.36700 | 0.21721 | 0.02803 | 0.08550 | 0.02750 | 0.01568 | 0.05064 | 0.01861 |
| DisenPOI [23] | 0.46741 | 0.60217 | 0.36294 | 0.24735 | 0.37451 | 0.21914 | 0.02898 | 0.08636 | 0.02869 | 0.01660 | 0.05133 | 0.01962 |
| GETNext [39] | 0.47973 | 0.63973 | 0.38436 | 0.27876 | 0.40645 | 0.24194 | 0.03167 | 0.09314 | 0.03325 | 0.01741 | 0.05508 | 0.02158 |
| MMPOI | **0.51410** | **0.71004** | **0.42352** | **0.30539** | **0.44092** | **0.26199** | **0.03334** | **0.10001** | **0.03743** | **0.01821** | **0.06038** | **0.02379** |

**Table 3: The recommendation performance of MMPOI and its variants on four datasets. Bold text indicates the best performance, while a wavy line represents the lowest performance.**

| Methods | NYC | | TKY | | New_orleans | | Philadelphia | |
|---|---|---|---|---|---|---|---|---|
| | HR@20 | NDCG@20 | HR@20 | NDCG@20 | HR@20 | NDCG@20 | HR@20 | NDCG@20 |
| MMPOI | **0.710037** | **0.423524** | **0.440923** | **0.261991** | **0.100015** | **0.037428** | **0.060380** | **0.023789** |
| MMPOI$_{w/o}I2T$ | 0.701595 | 0.418959 | 0.430189 | 0.250883 | 0.090528 | 0.034913 | 0.058113 | 0.023099 |
| MMPOI$_{w/o}MM$ | 0.688130 | 0.409545 | 0.417135 | 0.245984 | 0.085437 | 0.033078 | 0.052038 | 0.021044 |
| MMPOI$_{w/o}Meta$ | 0.693301 | 0.412886 | 0.414959 | 0.239766 | 0.089957 | 0.034433 | 0.056757 | 0.022714 |
| MMPOI$_{w/o}Geo$ | 0.684234 | 0.400607 | 0.394583 | 0.237148 | 0.087709 | 0.034067 | 0.054538 | 0.021687 |
| MMPOI$_{w/o}Cat$ | 0.686919 | 0.403356 | 0.409613 | 0.241524 | 0.093593 | 0.035867 | 0.057789 | 0.022297 |

adaptive multi-task learning, our model has relatively fewer hyper-parameters to adjust. We perform a grid search on the learning rate in {1e-1, 1e-2, 1e-3, 1e-4, 1e-5, 1e-6}. The other optimal hyper-parameters are also determined through grid searches and reported in the experiment analysis. We fixed the number of layers for both GCN and Transformer to 2. All models are evaluated on a Tesla V100 32G GPU card.

## 4.2 Performance Comparison

The performance of various recommendation methods on all four datasets is summarized in Table 2. In the table, the performance of MMPOI is highlighted in bold, while the best-performing baseline is underlined. From the table, we have the following observations:

Firstly, the proposed MMPOI model significantly outperforms the state-of-the-art sequential recommendation, multi-modal rec-ommendation, and POI recommendation methods on each dataset. Specifically, MMPOI improves the best baseline in terms of HR@20 by 11%, 8.5%, 7.4%, and 9.6% in NYC, TKY, New_orleans, and Philadel-phia, respectively. We credit the performance improvement to the effective integration of multi-modal content information of POIs with the check-in sequence and the application of our adaptive multi-task Transformer. Furthermore, the majority of recently pro-posed GCN-based POI recommendation algorithms perform better than those only using self-attention. This highlights the effective-ness of graph techniques in modeling relationships across multiple sequences.

Secondly, considering that most POI recommendation algorithms follow the sequential recommendation paradigm. Consequently, in our experiments, two advanced sequential recommendation al-gorithms (SASRec and ICLR) are selected as baselines. The exper-imental results indicate that directly applying sequential recom-mendation algorithms to the POI recommendation scenarios does

not yield satisfactory performance. The underlying reason is that sequential recommendation algorithms do not consider the con-textual information in the POI recommendation scenario, such as visiting time, spatial location, and POI category. This highlights the significance of considering the contextual information of check-in sequences in POI recommendations.

Thirdly, two state-of-the-art multi-modal recommendation al-gorithms (LATTICE and MMSSL) are selected as baseline models. These methods aim to improve item and user representations by modeling multi-modal content information. From the experimental results, it can be observed that the effectiveness of the multi-modal recommendation methods is inferior to that of the POI recommen-dation algorithms. Nonetheless, the multi-modal recommendation methods still exhibit certain effectiveness in the execution of the next POI recommendation tasks. These experiments emphasize the importance of integrating multi-modal content information when making POI recommendations.

## 4.3 Ablation Study

In this section, we design five variants of our method to justify the significance of key components in MMPOI: 1) MMPOI$_{w/o}I2T$: It uses the pre-trained image2text model to embed images directly, without converting them into textual descriptions. 2) MMPOI$_{w/o}MM$: It does not consider the multi-modal structural relationships of POIs when learning the POI representations. In other words, it only uti-lizes the user trajectory graph to learn the POI representations. 3) MMPOI$_{w/o}Meta$: It performs one-hot encoding for categories as opposed to embedding them with textual semantics, then uti-lizes them to initialize the nodes in the multi-modal trajectory flow graph. 4) MMPOI$_{w/o}Geo$ and 5) MMPOI$_{w/o}Cat$: They remove the auxiliary prediction tasks for grid cell and category in adaptive multi-task learning, respectively.

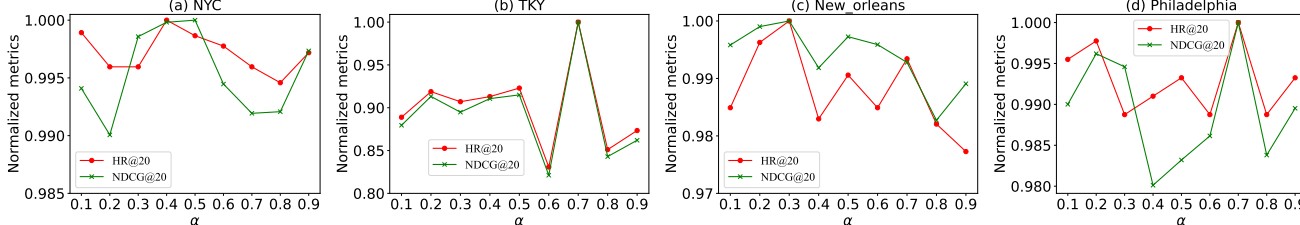

**Figure 2: Effects of $\alpha$ on the four datasets, where Normalized HR@20 (or NDCG@20) is obtained by dividing each HR@20 (or NDCG@20) to the maximum value of that metric.**

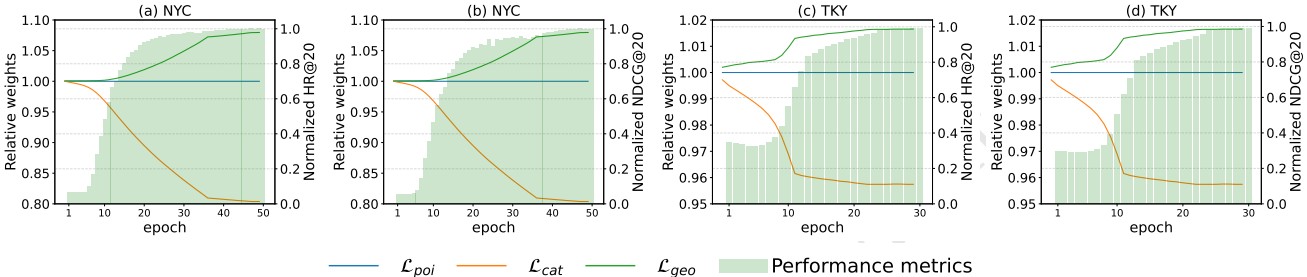

**Figure 3: Effects of the adaptive multi-task learning on NYC (a-b) and TKY (c-d) datasets.**

Table 3 summarizes the recommendation performance of MM-POI and its variants on all four experimental datasets. As shown in Table 3, MMPOI consistently outperforms five variants, which verifies the effectiveness of each key component in MMPOI. Specifically, MMPOI$_{w/o}I2T$ and MMPOI$_{w/o}Meta$ both exhibit a certain performance decrease compared to the full model MMPOI, indicating the effectiveness of bridging modal semantic differences and modeling textual semantic features for POI recommendation. Additionally, the performance of MMPOI$_{w/o}MM$, which does not take into account the multi-modal structure graph, decreases sharply compared with MMPOI. It demonstrates the strength of our designed multi-modal trajectory flow graph and the effectiveness of considering the multi-modal content of POIs. Moreover, as indicated in Table 3, the importance of multi-modal content and adaptive multi-task learning varies with datasets. In the Yelp dataset, considering the structural features of multi-modal content has the most significant impact on model performance, while in the FourSquare dataset, adaptive auxiliary prediction tasks are the most crucial for the model.

### 4.4 Discussion

In this section, we investigate the sensitivity of the parameter $\alpha$ and analyze the effectiveness and convergence of adaptive multi-task learning. Sensitivity analysis for other hyper-parameters (e.g. $k$NN-$k$ and dropout ratio) will be provided in the appendix.

*4.4.1 Effect of the parameter $\alpha$.* As presented in Section 3.3.4, $\alpha$ plays an important role in our method, as it serves to integrate multi-modal structural relationships and check-in sequence relationships between POIs during the learning process of multi-modal POI representations. $\alpha$ is selected from the range of [0.1, 0.2, ..., 0.9]. As shown in Figure 2, optimal $\alpha$ values vary across different datasets. The evaluation results indicate the effectiveness of considering multi-modal content features of POIs in the next POI recommendations.

*4.4.2 Effect of adaptive multi-task learning.* The uncertainty-based adaptive multi-task learning strategy will tend to learn the easier tasks first. We initialize $\phi_1$, $\phi_2$, and $\phi_3$ to 0.5, and select the weight of the main task (i.e. the next POI prediction task) as a reference baseline. As demonstrated in Figure 3, the weight assigned to the grid cell prediction task consistently exceeds that of the category prediction task, implying that the grid cell prediction task is associated with lower uncertainty. Intuitively, user movement is constrained by geographical distance, resulting in relatively lower uncertainty and making it easier to predict users' spatial movement. Additionally, with the dynamic changes in multi-task weights, the model trains and converges rapidly. In conclusion, the experimental results align with intuitive expectations, and demonstrate the effectiveness of adaptive multi-task learning.

## 5 CONCLUSION

In this paper, we propose a novel multi-modal content-aware framework for POI recommendations, which introduces the multi-modal content information of POIs into the next POI recommendation algorithm, effectively alleviating the data sparsity issue. In particular, we fully exploit pre-trained models to bridge the semantic differences between multi-modal contents and employ $k$NN sparsification to filter out noise. Moreover, we construct a multi-modal trajectory flow graph to effectively integrate multi-modal structural features with check-in sequential features, and design an adaptive multi-task Transformer to model user movement patterns. Additionally, we have collected and released datasets that contain multi-modal content information for multi-modal POI recommendation tasks. Extensive experiments on these datasets demonstrate the effectiveness of our proposed method from various perspectives.

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

# A  EXPERIMENT BASELINES

To evaluate the performance of our proposed model MMPOI, we compare it with the following baseline methods which can be divided into three groups.

1) Multi-modal recommendation methods:

- **LATTICE** [43] learns item-item structures for each modality and aggregates multi-modal information to model the latent semantic features of items.
- **MMSSL** [33] employs contrastive learning to discover self-supervised signals and learns modality-aware user preference and cross-modal dependencies.

2) Sequential recommendation methods:

- **SASRec** [11] applies self-attention mechanisms to the sequential recommendation and adaptively considers previous items for prediction.
- **ICLR** [3] employs contrastive self-supervised learning to capture user intent for sequence recommendation.

3) POI recommendation methods:

- **LSTPM** [27] introduces a context-aware nonlocal network structure to capture users' long-term preferences, and design a geo-dilated RNN for modeling their short-term preferences.
- **STAN** [22] introduces a bi-attention architecture to explicitly learn the spatio-temporal correlations within user trajectories.
- **PLSPL** [36] considers both long-term and short-term user interests, incorporating contextual information such as categories and check-in time for the next POI prediction.
- **GETNext** [39] constructs a directed trajectory graph to represent the correlation between multiple check-in sequences, and applies GCN to learn the representations of POIs.
- **DisenPOI** [23] constructs dual graphs for sequential and geographical relationships, and utilizes contrastive learning to enhance POI recommendation.

# B  SENSITIVITY ANALYSIS

## B.1  Effect of $k$NN-$k$

To mitigate the impact of noise in the multi-modal content information, we employ a $k$-nearest neighbors ($k$NN) sparsification approach to pruning the multi-modal structural graph. In this subsection, we investigate the influence of $k$ by varying it from [1, 5, 25, 50, 100]. As we can see in Table 4, our method performs the best with $k = 5$ for all datasets. A small value of $k$ (e.g., $k$=1) introduces only a small amount of multi-modal features and cannot effectively address the data sparsity issue. Conversely, increasing $k$ leads to the introduction of more noise, resulting in a decrease in model performance. Experimental results indicate that introducing an appropriate amount of multi-modal features could improve the accuracy of POI recommendations.

## B.2  Effect of dropout ratio

We tune the dropout ratio in the GCN and Transformer from [0.1, 0.3, 0.5, 0.7, 0.9] to control the model's robustness to various influencing factors. Figure 4 shows the normalized performance achieved by MMPOI under various combinations of diverse dropout ratios for GCN and Transformer. We note that a smaller dropout

**Table 4: The impact of $k$NN-$k$ on recommendation performance in terms of HR@20.**

| Datasets | $k$NN-$k$ | | | | |
|---|---|---|---|---|---|
| | 1 | 5 | 25 | 50 | 100 |
| NYC | 0.70906 | **0.71004** | 0.70940 | 0.70781 | 0.70749 |
| TKY | 0.43320 | **0.44092** | 0.42618 | 0.38667 | 0.39317 |
| New_orleans | 0.09794 | **0.10001** | 0.09765 | 0.09794 | 0.09850 |
| Philadelphia | 0.06024 | **0.06038** | 0.05943 | 0.05957 | 0.06024 |

ratio of Transformer generally improves the accuracy of our proposed MMPOI. On the other hand, the optimum performance is obtained when the GCN's dropout ratio is set to 0.3.

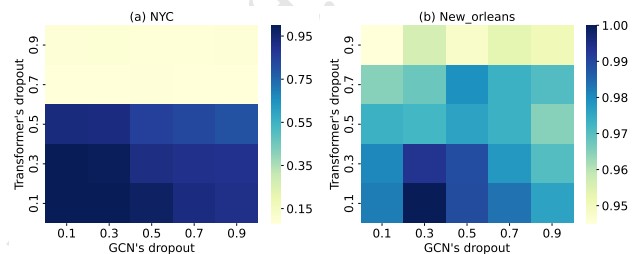

**Figure 4: The effect of dropout ratio in GCN and Transformer.**

