# OpenReview forum: "MMPOI: A Multi-Modal Content-Aware Framework for POI Recommendations"
_ACM.org/TheWebConf/2024/Conference — TheWebConf24_

### Official Review · Reviewer_HhLF · 2023-11-22

**Novelty:** 4
**Technical Quality:** 4

**Review:**

The paper introduces a multi-modal content-aware framework named MMPOI for location recommendations, addressing the challenge of data scarcity in existing systems by integrating multi-modal content information about locations. MMPOI utilizes pre-trained models for inter-modal conversion and employs a unified pre-trained model to extract modality-specific features, effectively bridging the semantic gap between different modalities. The framework also introduces a Multi-Modal Trajectory Flow Graph (MTFG) and an adaptive multi-task Transformer to model users' multi-modal movement patterns for location recommendation tasks.

## PROS:
1. The paper has a clear and straightforward structure, making it easily understandable.
2. The paper tries to address the data sparsity issue in POI recommendation by incorporating multi-modal content information, such as textual and image data, which is commonly associated with POIs, adding a novel dimension to the field of POI recommendation.
3. The paper identifies three challenges in applying multi-modal techniques to POI recommendation: The semantic space disparity among different modal contents in multi-modality, the substantial inclusion of noise in multi-modal data, and the notable semantic difference between multi-modal content and user check-in sequences.One notable strength of this paper is the systematic exploration of these issues, addressing each one individually and proposing corresponding solutions.

## CONS：
1. Lack of sufficient experiments or comparison with the latest baseline raises questions about the effectiveness of the method.
2. Insufficient discussion on limitations: The paper does not thoroughly explore the limitations and potential drawbacks of the proposed framework,such as computational complexity, scalability, or potential biases introduced by pre-trained models.
3. In addressing the significant noise issue in excluding multi-modal data, a cosine similarity function is employed for their modality-specific representation, retaining the edges with the highest similarity scores. However, the effectiveness of this approach requires further discussion.

**Questions:**

1. The paper mentions studies such as GETNEXT, AGRAN,STHGCN,etc., in the related work section. However, why were experiments only conducted with GETNEXT, the study published earliest in terms of release date, while experiments with the more recent AGRAN and STHGCN were overlooked?
2. The appendix discusses the impact of different values of K on addressing the noise issue. However, can more comprehensive reasons be provided in the main text to demonstrate the meaningfulness of using existing methods to address the noise problem?

**Reviewer Confidence:**

3: The reviewer is confident but not certain that the evaluation is correct

**Scope:**

3: The work is somewhat relevant to the Web and to the track, and is of narrow interest to a sub-community

---

### Official Review · Reviewer_sT3R · 2023-11-23

**Novelty:** 4
**Technical Quality:** 4

**Review:**

Summary:

This paper proposes a novel multi-modal framework to mitigate the data sparsity issue  for the next POI recommendation task through taking into account POI’s image and text information. The pre-trained models are used to align modal-specific features, which can be constructed into a multi-modal trajectory flow graph. Moreover, a geographic trajectory flow graph reflecting geographical dependencies between POIs and an adaptive multi-task Transformer are proposed to improve the model performance. Extensive experiments are conducted show the superiority of the proposed method.

Strengths:
1. This paper is well-constructed, with lucidity and straightforwardness. The rationale behind incorporating multi-modal information under the data sparsity issue is well-motivated.
2. A novel multi-modal framework for the next POI recommendation task is proposed, including the utilization of the pre-trained models, the construction of multi-modal graph, and the application of adaptive multi-task learning.
3. Extensive experiments are conducted show the superiority of the proposed method.

Weaknesses:
1. This paper releases a invalid code link, it’s unclear whether the authors are willing to offer all pre-processed multi-modal datasets rather than just a part of them.
2. The ablation experiment is not perfect. A variant MMPOI𝑤_{w/o}Geo&Geo graph should be conducted to further verify the effect of geographic trajectory flow graph. And a variant considering the randomly initialized time embeddings should be conducted to verify the effect of time2vec method.
3. The settings of some parameters is undefined, such as $d_m$ in line 331,  $l_m$ in line 596, and region number $V^G$. Moreover, the dimension of $D^G$ should be $\mathbb{R}_{V^G \times V^G}$.

**Questions:**

See weaknesses.

Other questions:

4. The authors should report the training resource and efficiency due to the utilization of pre-trained models. And the reason why the authors use BLIP2 and Sentence-Bert should be presented or compare the impact of different pre-trained models on next POI recommendation task.
5. In line 469, the POI category embeddings $E_C$ is used to represent the POIs and propagate the message in Equation (12), but the number of category should be far less than the number of POI. In others words, $E_C$ cannot reflect the uniqueness of each POI. Moreover, the specific number of category should be presented in Table 1.
6. Three specific loss calculation equations should be presented, and the reason why the authors use KL divergence to measure category prediction should be explained (Why not cross entropy loss?).
7. The paper studies the next POI recommendation problem, why do the authors title “for POI recommendation” instead of “for Next POI recommendation”?

**Reviewer Confidence:**

3: The reviewer is confident but not certain that the evaluation is correct

**Scope:**

4: The work is relevant to the Web and to the track, and is of broad interest to the community

---

### Official Review · Reviewer_n2K7 · 2023-11-23

**Novelty:** 6
**Technical Quality:** 5

**Review:**

The authors propose a novel multi-modal framework for POI recommendation to tackle data sparsity. The goal of this framework is to tackle the issue of data scarcity in POI recommendation by leveraging sequential information as well as POI information, coming from different modalities. The proposed architecture, namely MMPOI is made up of several components, which are designed to tackle three main challenges: (1) semantic heterogeneity of different modalities; (2) noise in multi-modal information; (3) semantic difference between multi-modal POI information and check-in information.

The extraction of POI embedding follows different phases: (1) a modality structure graph construction where each modality is represented by means of a graph with edges constructed based on modality-specific POIs similarities; each modality graph is then pruned based on kNN; (2) multi-modal Trajectory Flow Graph construction: the multi-modal graphs are combined with the user trajectory graph, combining multi-modal features and POI sequential relationships. (3) POI embedding: resulting POI notes are embedded by means of spectral GCN.

The extraction of check-in representations can be summarized as follows: (1) Geographic Trajectory Flow Graphs: the authors propose to include spatial information for the characterization of check-ins, based on the assumption that user preferences could reflect geographical co-occurrences of POIs; the resulting grid-cell nodes are embedded by mean of spectral GCN; (2) since time is strictly correlated to POI categories, the authors embed time by applying time2vector. (3) The final check-in representation is obtained by concatenating the user, time, grid-cell, and multi-modal embeddings.

The next check-in prediction is performed by means of a transformer model, which is trained on three joint tasks: next POI prediction, category prediction, and grid-cell prediction.

*Pros*:
- The work is clearly relevant to the track. Although its primary interest involves a niche sub-community, the proposed methodology can be partially generalized to other systems that could leverage multiple modalities for the enrichment of content information;
- The proposed methodology is certainly novel within the scope of POI recommendation systems;
- The design motivations behind each subsystem within the framework are well-established and substantially backed. The authors provide clear examples of real-world use cases in order to explain the technical backing of each solution further;
- The problem is well formalized and the manuscript readability is excellent, even for readers who may be unfamiliar with the topic;
*Cons*:
- The authors did not provide the source code for reproducibility, which in my opinion is crucial for an experimentally intensive work like this. While the methodology is adequately outlined and theoretically replicable, the lack of access to the code introduces the potential for discrepancies in results compared to those reported by the authors.
- The claim that noise can be considerably mitigated by means of a kNN-based sparsification of modality-specific graphs is a strong claim that necessitates more insight. I am not convinced by this claim particularly because the choice for which edges to keep is made based on the relative ranking of different POIs similarity scores. From my understanding, this could still lead to noisy edges in the presence of outliers. In addition, choosing a fixed k could lead to some degree of information loss in the resulting graph for highly connected POIs.
- The methodology section lacks more details on how the user-embedded representations are computed.
- The ablation results are interesting but could benefit from additional settings. For instance, it could be interesting to assess the performances when removing specific information entirely from either the check-in embeddings or the POI embeddings (e.g. no category information) to validate the authors' claims further;
- While the primary objective of the authors is to address data sparsity in POI recommendation systems, the extent to which this system effectively mitigates data sparsity remains unexplored. Further investigation may be necessary, especially in relation to the characteristics of the utilized datasets.
- For a fair comparison with other existing multi-modal recommendation systems, it might be beneficial to specify what kind of modalities they can consider.

**Questions:**

- Could you provide more details about how the user embeddings are obtained?
- Did you explore additional ablation configurations?

**Ethics Review Description:**

No issue

**Reviewer Confidence:**

3: The reviewer is confident but not certain that the evaluation is correct

**Scope:**

4: The work is relevant to the Web and to the track, and is of broad interest to the community

---

### Official Review · Reviewer_8umR · 2023-11-24

**Novelty:** 4
**Technical Quality:** 5

**Review:**

The research work utilizes available multi-modal information (both textual and visual) to predict the user’s next point of interest (POI). The reason for attempting to solve the problem is data sparsity as users only visit a few preferred places which hinders the task of recommendation of new POIs to the users. To effectively integrate the multi-modal features of POIs and the check-in features of users, they construct a denoised dense multimodal trajectory flow graph (MTFG) to learn the multi-modal representations of POIs. They conducted their experiments on 4 publicly available datasets: 2 from Yelp and two from FourSquare. And their experiments shows a performance gain of 8% to 11% on SoTA models.

Pros:

· Overall, decent work. Addresses one of the significant research topic as it does not provide just the convenient exploration of unfamiliar locales but also empowers businesses with sharpened marketing tactics

· The contribution is sufficient. Apart from technical part, they have also generated and released a dataset that contains multi-modal content information for POI recommendation tasks.

· The proposed work proposes the use of kNN sparsification to mitigate the impact of multi-modal noise on the accuracy of recommendations. This part is avoided by most of the recent works.

· The paper is easy to read and follow.

Cons:

· The paper didn’t take into account other few significant works in multi-modal recommendations such as
https://dl.acm.org/doi/abs/10.1145/3474085.3475709 and
https://dl.acm.org/doi/10.1145/3511808.3557387.

· A more accurate way of assessing the relevance or similarity could be used instead of cosine similarity.

I acknowledge that I have read the rebuttal.

**Questions:**

Please clarify the Cons points in the detailed review.

In addition:

· Is content-aware different than context aware? If yes, did you consider adding context to your model?

· How do you plan to mitigate the chances of error propagation given that the proposed model is a pipeline model?

**Reviewer Confidence:**

4: The reviewer is certain that the evaluation is correct and very familiar with the relevant literature

**Scope:**

3: The work is somewhat relevant to the Web and to the track, and is of narrow interest to a sub-community

---

### Official Review · Reviewer_PWd4 · 2023-12-02

**Novelty:** 5
**Technical Quality:** 4

**Review:**

This paper proposes a novel multi-modal content-aware framework for POI recommendations, which introduces the multi-modal content information of POIs into the next POI recommendation algorithm. Moreover, authors construct a multi-modal trajectory flow graph to effectively integrate multi-modal structural features with check-in sequential features, and design an adaptive multi-task transformer to model user movement patterns.  My detailed comments on the strong and weak points of this paper can be found below.

Strong points:
S1: It is important to consider multi-modal content information associated with POIs.
S2: This paper proposes a novel MMPOI model to address the issue of data sparsity by incorporating the multi-modal content information of POIs.
S3: MMPOI establishes a geographical sequence pattern and employs an adaptive multi-task Transformer to capture users’ comprehensive movement patterns.
S4: Experiments conducted on Foursquare and Yelp datasets show that MMPOI outperform state-of-art methods.

Weak points:
W1: The datasets are small. All of them have #User<4K and #POI<8K. There are some much larger datasets used in https://proceedings.mlr.press/v101/li19a.html.
W2: Time complexity analysis is needed. The computation costs of MMPOI are missed for both theorical analysis and experiments.
W3:  Multi-modal data contains a considerable volume of noise, which could introduce a significant quantity of noise into POI recommendation tasks (Second challenge referred in Sect1.Introduction). However, there is no experiment of robustness to noisy data.

**Questions:**

Q1: Mapping different modalities of data into same latent space has been studied before. What’s the difference between MMPOI and these methods?

Q2: The datasets contain some category information. MMPOI seems to use this information during the training phrase. As noted in [3], it only utilizes these attributes to study the effectiveness of the proposed ICLR both quantitatively and qualitatively. Is it fair to compare these methods that do not use category information with MMPOI?

Q3: Why there remains virtually unchanged when k increases?  Although MMPOI performs best with k=5 for all datasets, the differences of k=[1,5,25,50,100] are narrow in Table 4. Authors say increasing k leads to the introduction of more noise, resulting in a decrease in model performance. However, k=100 outperforms k=50 (also, k=50 outperforms k=25) in some datasets.

**Reviewer Confidence:**

3: The reviewer is confident but not certain that the evaluation is correct

**Scope:**

3: The work is somewhat relevant to the Web and to the track, and is of narrow interest to a sub-community

---

### Decision · Program_Chairs · 2024-01-22

**Decision:**

Accept

**Comment:**

The paper presents an approach to address data sparsity in POI recommendations by integrating multi-modal content. The reviewers recognize the strengths of the paper for its novelty, clear structure, and contribution to addressing the challenge of data sparsity in POI recommendations.

 However, there are notable weaknesses such as small dataset size, absence of time complexity analysis, lack of robustness to noisy data, and omission of comparison with significant works in the field. Additionally, the absence of source code for reproducibility (seems solved during rebuttal) and insufficient exploration of the method's effectiveness in mitigating data sparsity are concerns.

 The paper is technically sound but would benefit from addressing these issues to strengthen its quality.